# Rhythm of the Night (and Day): Predictive Metabolic Modeling of Diurnal Growth in *Chlamydomonas*

Alex J. Metcalf,[a] Nanette R. Boyle[a]

aChemical and Biological Engineering, Colorado School of Mines, Golden, Colorado, USA

**ABSTRACT** Economical production of photosynthetic organisms requires the use of natural day/night cycles. These induce strong circadian rhythms that lead to transient changes in the cells, requiring complex modeling to capture. In this study, we coupled times series transcriptomic data from the model green alga *Chlamydomonas reinhardtii* to a metabolic model of the same organism in order to develop the first transient metabolic model for diurnal growth of algae capable of predicting phenotype from genotype. We first transformed a set of discrete transcriptomic measurements (D. Strenkert, S. Schmollinger, S. D. Gallaher, P. A. Salomé, et al., Proc Natl Acad Sci U S A 116:2374–2383, 2019, https://doi.org/10.1073/pnas.1815238116) into continuous curves, producing a complete database of the cell's transcriptome that can be interrogated at any time point. We also decoupled the standard biomass formation equation to allow different components of biomass to be synthesized at different times of the day. The resulting model was able to predict qualitative phenotypical outcomes of a starchless mutant. We also extended this approach to simulate all single-knockout mutants and identified potential targets for rational engineering efforts to increase productivity. This model enables us to evaluate the impact of genetic and environmental changes on the growth, biomass composition, and intracellular fluxes for diurnal growth.

**IMPORTANCE** We have developed the first transient metabolic model for diurnal growth of algae based on experimental data and capable of predicting phenotype from genotype. This model enables us to evaluate the impact of genetic and environmental changes on the growth, biomass composition and intracellular fluxes of the model green alga, *Chlamydomonas reinhardtii*. The availability of this model will enable faster and more efficient design of cells for production of fuels, chemicals, and pharmaceuticals.

**KEYWORDS** transcriptomics, systems biology, algae, diurnal light, metabolic modeling, computational biology, computer modeling, mathematical modeling, metabolic engineering, metabolism

The dawn of a new century has also led to an awakening about our energy use; there has been a concerted push to develop more sustainable sources of energy and feedstock chemicals. One of the major challenges in developing more sustainable fuels and chemical supply chains is that the current industry is large; significant capital has already been spent to develop the infrastructure for processing and distribution. To avoid an extensive and expensive redesign of infrastructure, renewables will have to be chemically similar to the fossil fuels we currently rely on. The fossil fuels we use today are derived from prehistoric biomass; therefore, it is a logical extension to engineer biomass to produce high concentrations and/or yields of equivalent chemical feedstocks and fuels that can be "dropped in" to current infrastructure. This approach has been successfully commercialized for a few chemicals, such as 1,3-propanediol (1, 2), 1,4-butanediol (3–6), and succinate (7) (among others). However, one drawback of these commercialized ventures is that they utilize heterotrophic bacteria, which require a source of reduced carbon which is derived from corn or sugar cane.

Address correspondence to Nanette R. Boyle, nboyle@mines.edu.

The authors declare no conflict of interest.

A more sustainable approach is the use of photosynthetic organisms that can grow on $CO_2$; unfortunately, the development of photosynthetically derived fuels and chemicals has fallen far short of the touted potential because the unique demands of photosynthetic organisms have not been considered in the strain development process. For example, algae and cyanobacteria have extremely strong circadian rhythms (8–10) which control the expression of cellular transcripts and proteins and metabolic fluxes (9, 11, 12). Additionally, they are strongly entrained: cells that are removed from light/dark cycles will continue to display circadian behaviors for days afterward (13). These rhythms have a profound impact on the cell but unfortunately have not been studied well in the context of how they impact engineering efforts. Currently, almost all metabolic engineering efforts in algae and cyanobacteria rely on growth in laboratory conditions with a continuous supply of light (14–27). This results in a steady-state growth environment that more closely mimics that of heterotrophic bacteria and enables more straightforward design and engineering of cells. However, large-scale growth of photosynthetic organisms necessitates growth under diurnal conditions outdoors, and the strong circadian rhythms that lead to dynamic gene expression can confound engineering efforts, as was reported by Cheah et al. (15). This also means that engineering strategies that have been shown to result in increased productivity under lab conditions will not directly translate to increased productivity in diurnal growth and is one reason why most current industrial algal production uses wild-type strains (28). Therefore, it is imperative that we develop tools that will enable more predictive and rational engineering of algal cells in diurnal growth.

Metabolic modeling techniques (especially constraint-based stoichiometric models) have proven extremely useful in minimizing the time it takes to develop heterotrophic production strains (29–32). One of the main assumptions in stoichiometric modeling approaches is steady state; this means that metabolite pool sizes are assumed to stay constant. While this approach works well for heterotrophic bacteria in exponential growth, it cannot be used to model the transient nature of photosynthetic growth in day/night cycles. When grown in diurnal light, 30 to 50% of genes show cyclic expression in *Arabidopsis thaliana* (33, 34) and up to 80% of genes in cyanobacteria, diatoms, and algae show periodic expression (35–38). These changes in gene expression impact metabolic fluxes, so to build a predictive model of photosynthetic metabolism, changing gene expression must be integrated into the model. Circadian rhythms do not affect only the cell's fluxes: the actual biomass composition of the cell changes over the course of the day as well, and photosynthetically grown cells store starch or glycogen during the day to support metabolism at night. This also requires a different formulation of stoichiometric modeling as the steady-state assumption necessitates the use of a static biomass formation equation, but diurnally grown cells alter their biomass composition significantly from day to night. The unique properties of diurnal growth in photosynthetic cells means that a new modeling framework must be developed in order to be able to more accurately predict metabolism.

Here, we describe the development of a transient modeling approach capable of predicting phenotype from genotype in the model green alga *Chlamydomonas reinhardtii*. This model combines the easy-to-implement stoichiometric constraint-based model of *C. reinhardtii* (39) with experimentally determined transient genetic constraints on reaction bounds and a decoupled biomass formation equation that enables flexibility in the production of biomass components. By incorporating the additional complexity to account for the transient nature of diurnal growth, we are able to increase the predictive nature of our model and simulate intracellular fluxes, biomass composition, and growth in diurnal light.

## RESULTS AND DISCUSSION

**Fitting and clustering of transcriptomic profiles.** Prior to using the experimental data for constraining the model, they first have to be converted from discrete data to continuous so that we can model smaller time steps than the experimental data (2-h

time steps). This way, the abundance of any transcript can be modeled at any time step. By fitting transcripts to best-fit functions, we can also easily compare the profile of expression across several genes (Fig. 1). The transcript expression profiles are clustered with those of other transcripts of similar expression patterns. We have chosen to highlight 4 specific genes and the most closely associated expression profiles in Fig. 1. As previously discussed, up to 80% of genes in cyanobacteria, diatoms and alga show periodic expression (35–38) in diurnal light; this makes it difficult to identify which housekeeping genes to normalize gene expression to. Therefore, the first gene we investigated further was *RACK1* (*CBLP*), which is commonly used as a housekeeping gene (40) and which shows very little variation in expression over time. We also found 19 named genes that are clustered closely with *RACK1*; these may be good candidates for housekeeping genes (see Data Set S1 in the supplemental material). We noticed that the expression profiles of many genes were most accurately modeled by the Kronecker Delta function; when we investigated this cluster, we saw that they clustered with known stress response genes, such as *LHCSR3.1* (41) (see Data Set S1). This is likely more an artifact of the experimental design, where the light is turned on immediately at the onset of day instead of ramping up as it would do in natural conditions. It is well known that ribulose-1,5-bisphosphatase carboxylase-oxygenase (RuBisCO) is a light-regulated enzyme (42); our expression profile of *RBCS1*, the gene encoding the small subunit of RuBisCO, shows that the cell anticipates the onset of light by starting transcription of this gene prior to the onset of light and the peak transcript abundance occurs 6 h after the onset of light. This profile is also clustered with multiple other named genes, including *FUO1*, associated with the electron transport chain, and *PGK1*, which encodes phosphoglycerate kinase, a critical enzyme within the Calvin cycle. (see Data Set S1 for more details). We also looked into genes known to be expressed at night, such as *FAP85*, a gene encoding a flagellar protein; genes associated with the synthesis of cilia are coordinately expressed at night following cell division (43). Other genes closely clustered with *FAP85* include *CPC1*, which is associated with the central pair microtubule complex (44), as well as 10 other genes specifically identified as encoding flagellum-associated proteins (see Data Set S1). Fitting transcript profiles to continuous functions not only aids in the development of transient constraints for the model but also helps to identify similar expression profiles and potentially gene function.

**Dynamic diel modeling.** By dynamically adjusting the model constraints according to transcriptomic abundance, we constructed a metabolic model that is capable of accurately describing the shifts in metabolism due to circadian rhythms and changing light conditions. In our simulations, we had an on/off onset of light, but due to the addition of gene expression constraints, we saw a ramp-up of carbon flux though the Calvin-Benson-Bassham cycle over the course of the day; it peaked at 5 h, after which carbon flux was ramped back down as the simulation moved toward the onset of night (animated in Movie S1). Despite the simulated cell experiencing the same amount of light at both time points, it utilizes less of it early in the day because the transcripts associated with the light reactions are less present, thereby reflecting transient physical limitations within the constraints of a dynamic model. These daytime ramps are driven by changes in expression of the photosystems and associated photosynthetic light reactions, but they end up reflected in the carbon flux throughout the cell as well (see external supplemental figures at https://github.com/metcalex/Transcriptomic _Circadian_Modeling_Supplemental/tree/main/Additional_Images). This ramping supports that the instantaneous light switch likely has a somewhat minimal overall effect on the cell's phenotype and growth relative to the gradual rise and fall in illumination the cell would experience under natural light. While some stress transcripts are produced at the moment of illumination, the transcript ramps associated with energy production still combine to produce light uptake bounds that rise and fall over the course of the day, reflecting the natural ramp of outdoor light. This consistency implies that the immediate light switch was not particularly impactful on energy availability; if it were, we would expect to see that light harvesting transcripts become immediately

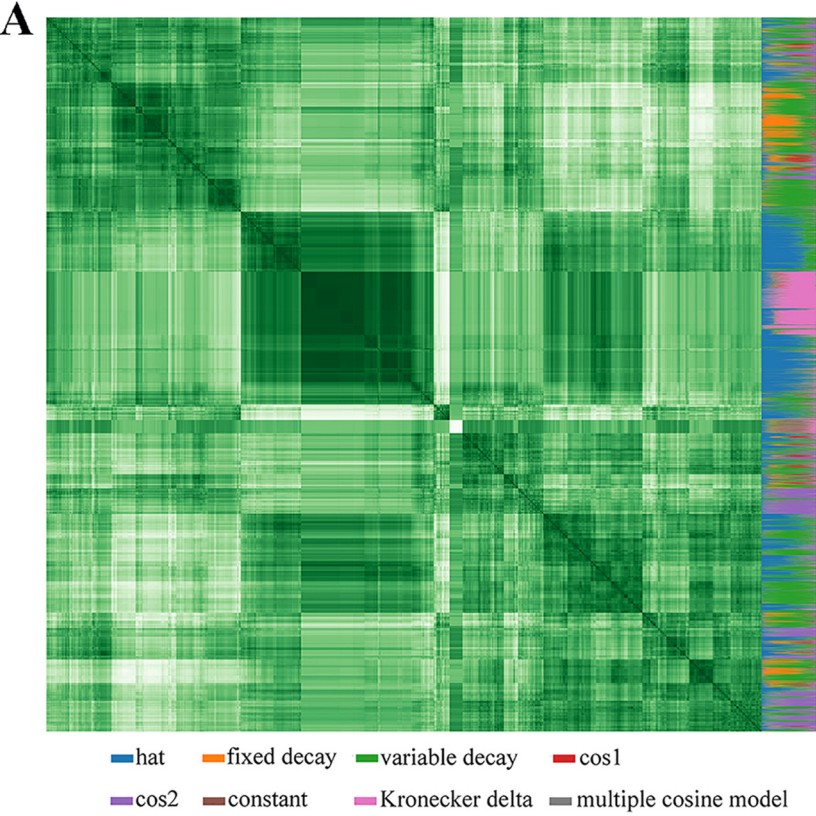

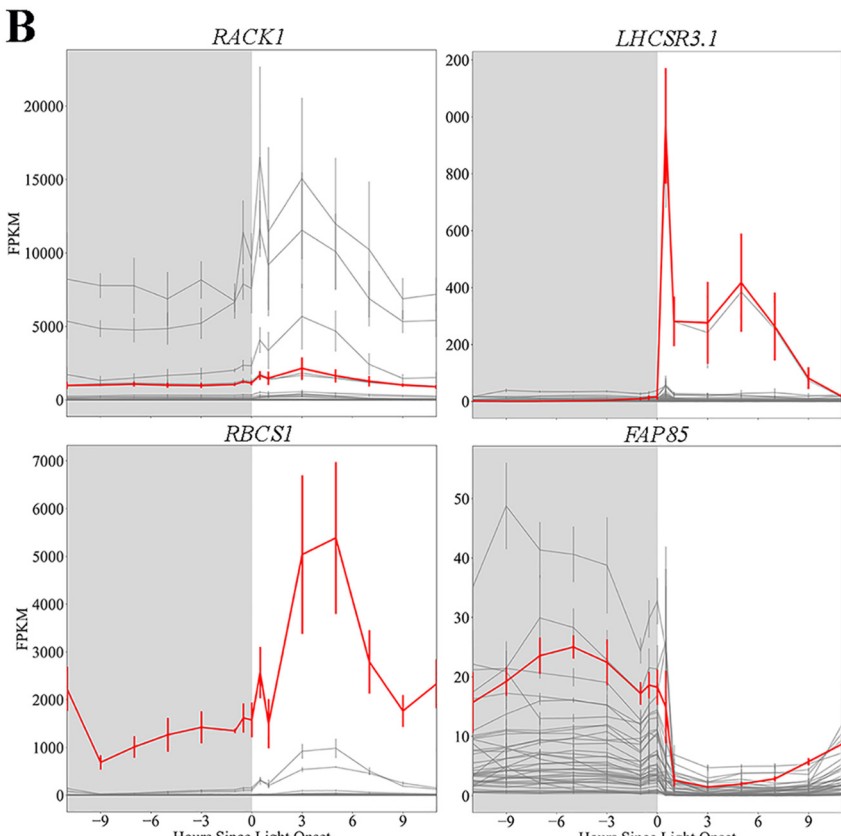

**FIG 1** Clustering time-dependent transcriptomic data provides insight into gene expression patterns. (A) Heat map of all expression patterns within the data set. Every pixel in the main square of this

expressed at full force and persist for the length of the day. As light uptake is the limiting overall factor for autotrophic growth under non-nutrient-limited conditions, the rise and fall imply that the cell's growth is not drastically affected by the sudden onset of light. However, it is possible that the internal constraints on the cell's metabolism are shifted by the onset; that is, the cell still experiences roughly the same ramp up and down of light harvesting that it would under natural light onset, but—due to changes in the expression patterns of other transcripts associated with other subsets of metabolism—the cell allocates the energy captured from the light it receives differently. This change would result in the cell producing different components of biomass at different time points, but the overall effect on total biomass would likely be relatively minimal, as the ramp in light uptake still remains; the cell would just have a slightly different composition at different time points. We therefore remain confident that the sudden light onset does not invalidate overall predictions of this model. We also observed the cell shuttling large amounts of carbon between the chloroplast and the mitochondria, likely to make use of the more efficient mitochondria ATP synthase (45). While this magnitude of transport could occur during photosynthetic growth, it is also possibly just an artifact of constraint-based modeling. Because constraint-based modeling uses optimization to calculate flux vectors, if an inefficient reaction is not strictly required, the model will avoid allocating flux through it. This phenomenon can also be seen in the model's lack of photorespiration and photoinhibition. As these processes consume energy without increasing biomass and are not enforced by constraints, the model will invariably allocate no flux to them. Verifying the magnitudes of such inefficiencies and adding in modeling constraints to account for them are areas of potential future investigation that could increase the accuracy of the model under certain conditions where such processes may predominate. Two potential conditions of interest on this front are high light or low carbon availability, as these conditions encourage photoinhibition and photorespiration, respectively—but because these conditions are not encountered in either the model's source data set or the simulated environments, we believe that the model still captures a significant portion of the cell's metabolic processes. Finally, the cell experiences a night slowdown, during which time it simply attempts to produce the maintenance ATP and does not encounter any significant reaction bounds. This phenomenon is supported by the finding of Strenkert et al. that the cell does not utilize its full respiratory capacity at night (11). Figure 2 shows flux maps of carbon metabolism and light harvesting reactions over a 24-h period with 12 h of light and 12 h of dark.

**Modeling mutant phenotypes.** The model is capable of predicting phenotype based solely on genomic and transcriptomic data. As validation, we simulated the phenotype of a starchless mutant (*sta6*) (46) in diurnal light both with acetate and on minimal media and a wild-type strain under the same conditions. Based on our simulations, the wild-type strain can grow in diurnal light in both conditions, though as expected, in the presence of a reduced carbon source (acetate), growth rate is higher. However, the *sta6* mutant is capable of growth only in the presence of acetate: as expected, the model predicts that a cell incapable of storing starch during the day cannot sustain itself at night, a result that is in agreement with data published by Davey et al. (47). This also highlights the power of our modeling approach, because a normal flux balance analysis (FBA) model would not predict that this gene is essential. Due to the way

**FIG 1** Legend (Continued)
image represents the degree of closeness between a pair of best-fit transcript functions, while the column on the right side indicates the relative weight of each of the best-fit models to the transcript in question; the larger the amount of a given color in the column, the higher the probability that that model fits the actual expression pattern better than any other of the evaluated choices. The multicolored column on the right of the heat map indicates the weight of each proposed curve fit to the gene expression pattern for a given gene; for example, if the majority of the column is pink, then that gene is best fit by the Kronecker Delta function. (B) Examples of selected genes of interest (shown in red) and genes which are most closely coexpressed (dark gray lines). The genes included in each cluster shown above are provided in Data Set S1. FPKM, fragments per kilobase per million.

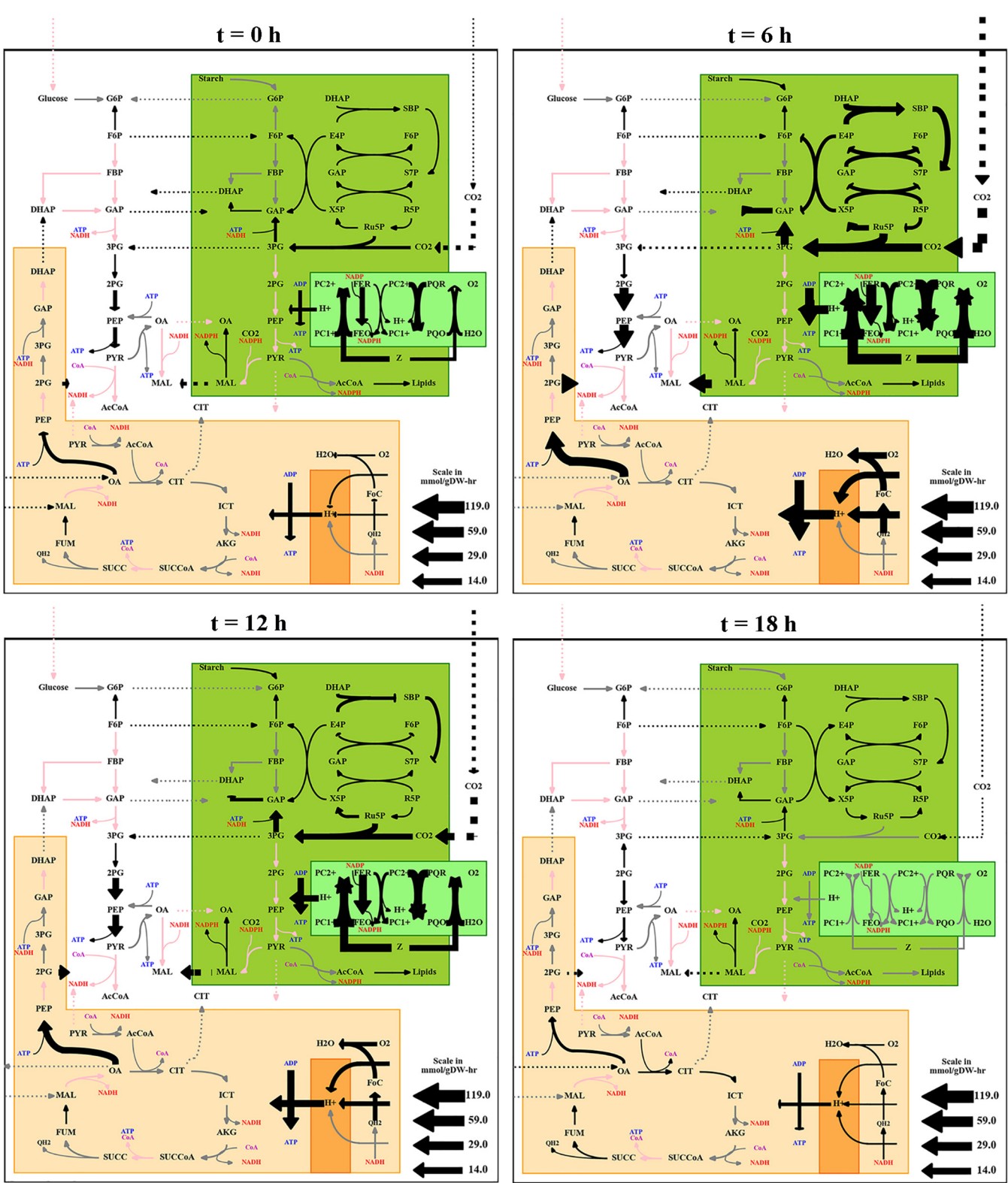

**FIG 2** Flux maps of central carbon metabolism and light harvesting reactions over a 24-h period with 12-h light and 12-h dark conditions. Labels above the flux maps give time since the onset of light; at *t* of 12 h, the light turns off. The thickness of the arrows indicates the amount of flux in millimoles per gram (dry weight) per hour through the given reaction, and the color indicates inclusion: black lines carry significant flux, gray lines do not, and red lines are not in the model of this organism. The model includes compartmentalization of reactions as shown: orange represents the mitochondria and green represents the plastid; dotted lines indicate transport flux between these different compartments. It is clear from these selected time points that the cell experiences dramatic shifts in carbon and light reactions during the day. For abbreviations, see Text S2 in the supplemental material.

FBA is formulated, it can model only steady-state conditions and not the transient changes that occur during the switch back and forth. This can result in knockouts that are not fatal under steady state becoming so, such as for Cre02.g145800.t1.2, a gene that encodes mitochondrial malate dehydrogenase. As might be expected from a cell that heavily relies upon shuttling carbon to the mitochondrial ATP synthase, removing the cell's ability to do so efficiently results in long-term failure to thrive. There are also some knockouts that are predicted to overproduce in a circadian environment, despite being fatal in a steady-state one, but these are universally the result of the dynamic model changing biomass constraints. Because the model does not strictly constrain the ratios of biomass components, the model can suggest that some knockouts are feasible despite their inability to produce biomass components; a good example of this is Cre17.g728950.t1.1, which controls production of flavin adenine dinucleotide (FAD). This redox coenzyme is only required in small amounts in the biomass equation, but it is large and therefore metabolically costly to produce. Mutants that cannot make it are predicted to grow faster, but this result is unlikely to be experimentally borne out. A full summary of these knockouts is available in Data Set S1.

Our model can predict not only growth and metabolic fluxes but also changes in biomass composition. This feature is incredibly important, as most efforts to engineer algae for biofuels have focused on the accumulation of lipids or fatty acids and our model is capable of predicting the effect of environmental or genetic changes on biomass. We used the model to predict the effect of nitrogen limitation on the accumulation of lipids in the wild type and *sta6* mutant in both continuous light and diurnal light. In continuous light, both strains accumulate more lipids with decreasing nitrogen in the medium. We also saw the phenotype that has been reported in a number of papers on *sta6* (25, 46, 48–50), one capable of accumulating more lipid than the wild type. When we repeated the simulations for growth in diurnal light, however, the wild-type strain lost its ability to respond to nitrogen stress with the accumulation of lipids. Regardless of the nitrogen limitation imposed on the model, the lipid content remained nearly constant. As we described earlier, the *sta6* mutant is incapable of growth in diurnal light; its failure to thrive is repeated (and likely exacerbated) with nitrogen limitation. This feature of the model lets us evaluate how external or internal changes in the cell impact biomass composition.

**Genotype-to-phenotype prediction.** The model was also used to predict phenotypes of other genetic knockouts (Fig. 3C). A key parameter for economical production of algal based biofuels is productivity; for growth-associated products, higher growth rates lead to higher productivities. Therefore, we first focused on the impact these gene knockouts had on growth. We designed a simulation that resulted in one doubling over the course of 1 week for the wild-type strain, in order to place a heavy energetic load on the cells, and then ran every knockout under the same conditions. Because of the experimental setup, any knockouts that impair growth have a normalized biomass less than 2, while those that result in higher biomass have a normalized biomass larger than 2 (Fig. 4). Not surprisingly, many of the underperforming mutants are knockouts of critical enzymes involved in pigment biosynthesis, chlorophyll biosynthesis, and light harvesting. Data Set S1 provides more detail for all mutants with a growth defect of 25% or more; of the 146 knockouts in the table, 23 encode enzymes which are part of the GreenCut2 (51), a functional classification of enzymes specific to the plant lineage. We investigated the functionality of these genes with Phytozome (52) and then compared these mutants to the genome-wide knockout library constructed and tested for growth rate in minimal medium photoautotrophically by Li et al.; given the major growth defects predicted by the model, it is not surprising that most were not found (53). Of the 1,355 genes in the model, only 39 were predicted to result in a 10% or higher increase in biomass (see Data Set S1). The simulation predicts higher growth in diurnal conditions, but closer inspection of the list can throw out several possibilities prior to constructing the mutant. For example, *nic2*, *nic7*, and *nic13* are auxotrophic for nicotinamide (54), two strains are deficient in genes encoding enzymes present

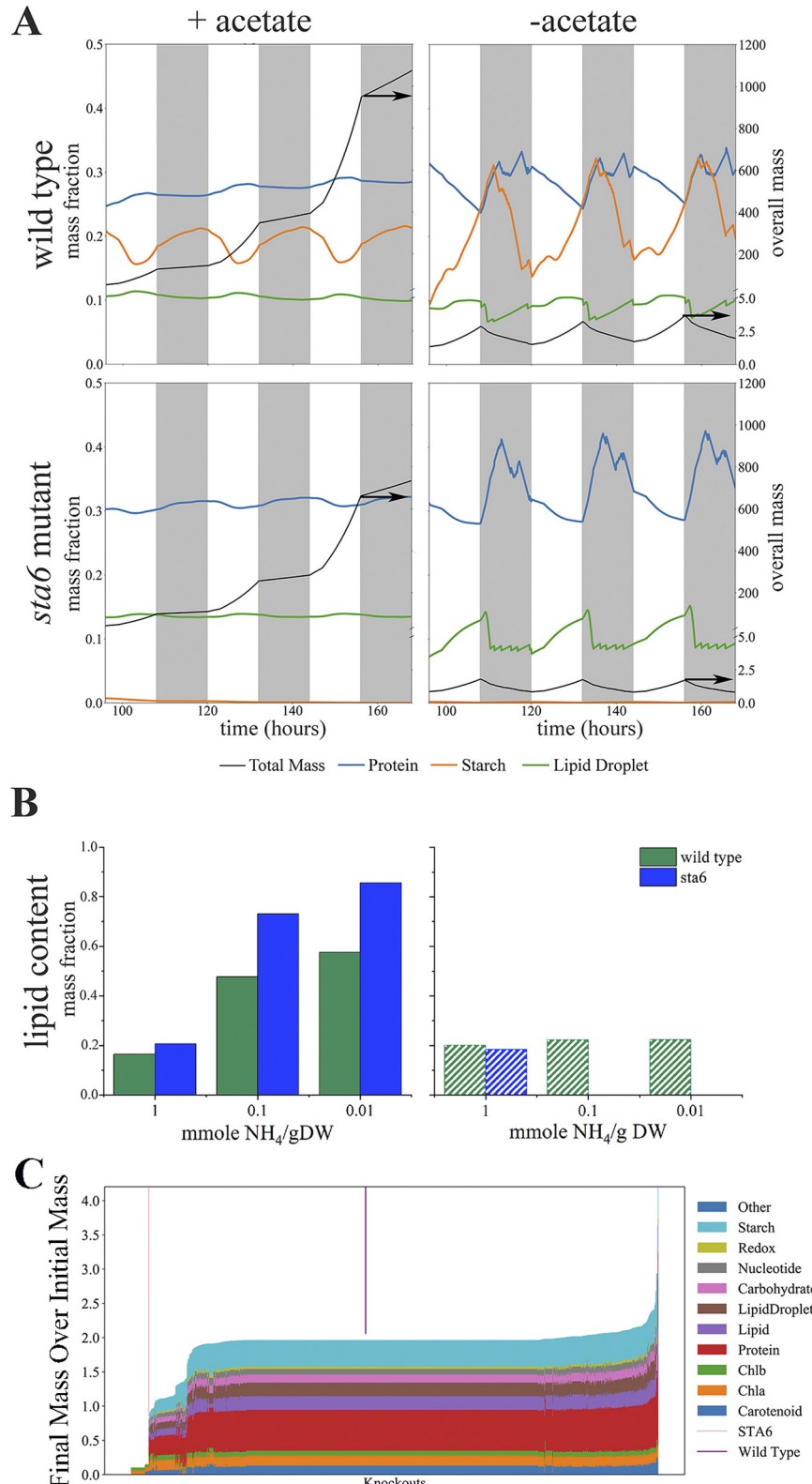

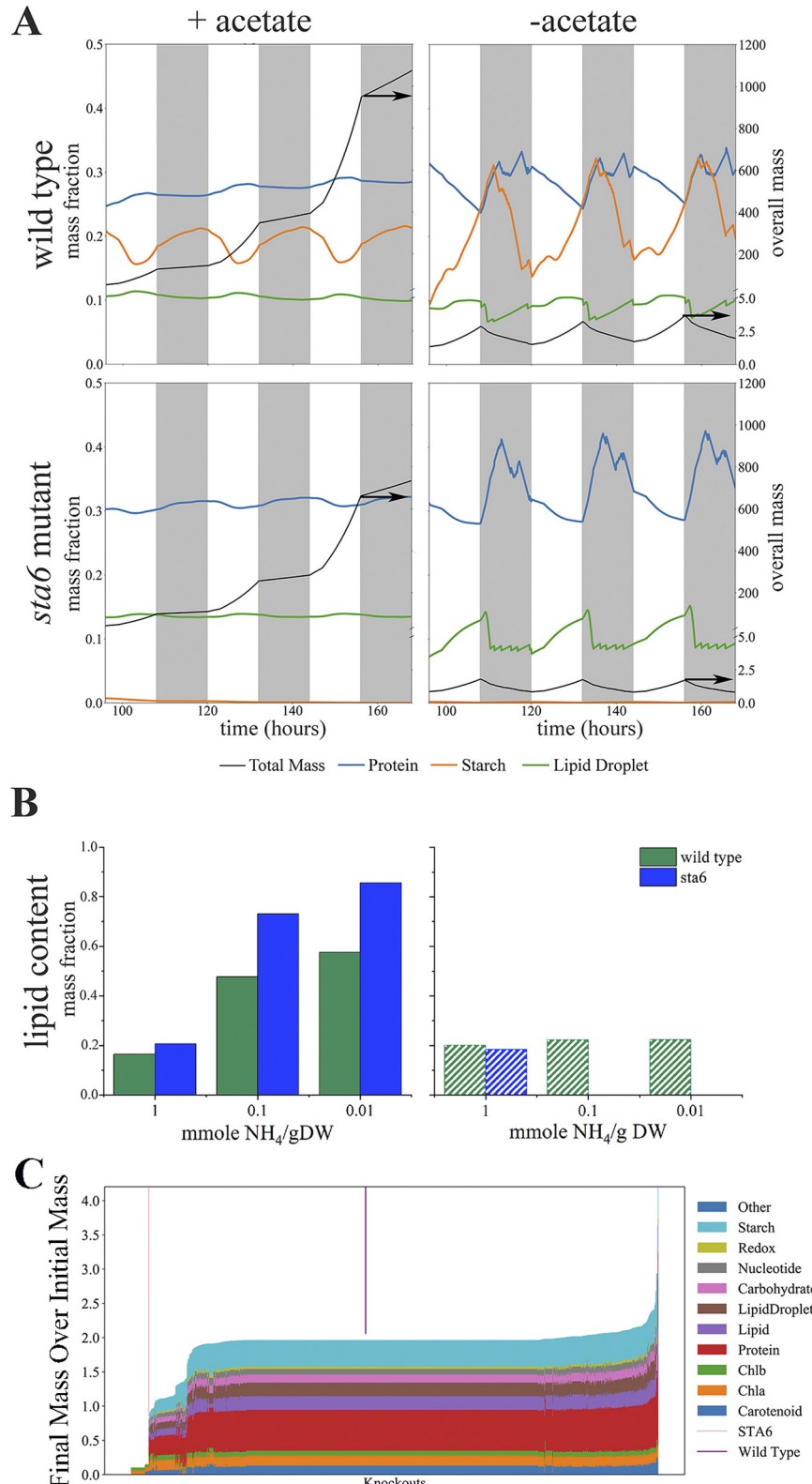

**FIG 3** Predicting phenotype from genotype. (A) Growth and biomass composition simulations for the wild type (WT) and *sta6* strain in diurnal light (the lighter sections of the plots represent the illuminated phase, while the darker sections represent darkness) with and without an organic carbon source; mass is normalized and therefore unitless. In the presence of acetate, both strains were able to sustain growth as indicated by an increase in mass over time, with the wild-type strain

in the photorespiratory pathway (*GYD1* and *AGT2*), thus requiring high $CO_2$ for growth, and one strain is lacking a key glyoxylate shunt enzyme (*MAS1*), which is a known carbon-conserving mechanism. The other mutants predicted to have better growth have not yet been characterized experimentally for growth in diurnal light.

**Conclusion.** While steady-state metabolic modeling is an important tool for biological engineering, photoautotrophic species exhibit strong circadian rhythms that are evident in the cyclic changes in transcript abundance and biomass composition. These transient changes require a modeling approach that allows for the dynamic changes occurring in the cell. In this study, we leveraged the ease of implementation of dynamic FBA models with gene expression and biomass composition data to develop a model that is capable of predicting phenotype in diurnal light. This model simulates growth in both day and night continuously, accounting for catabolism, changing objective functions, and transcriptomic constraints. We have shown its ability to accurately predict phenotype from genotype, making it a useful tool for both physiological interrogation of biology and metabolic engineering efforts. When that predictive power is combined with the inherently parallelizable nature of *in silico* modeling, it is possible to quickly assess bioengineering approaches for feasibility, as demonstrated by individual simulations of a broad swath of single knockout mutants. Additionally, this approach requires relatively few discrete types of data: the primary requirements are only a preexisting constraint-based model, an appropriate transcriptomic data set, and a time course of biomass. There is no other organism-specific information required. Because of this, our approach can be generalized to any other photosynthetic alga species with similar data availability.

## MATERIALS AND METHODS

**Informational pipeline.** In order to utilize a large data set of circadian transcriptomic expression patterns, we built a pipeline for mass fitting simple expression curves to every single individual transcript. These curves are simple enough to reduce overfitting, but they also have enough parameters to preserve critical cyclical information. Once these curves were fit, we used them in two ways. The first was a clustering approach that enabled us to assess how every single transcript's expression pattern compared with every other transcript within the cell. This produced a heat map where several clear correlations could be identified, including the collection of genes with a strong response to sudden light onset. The second was the integration of these curves into a genome-scale metabolic model; by querying the transcriptomic curves at every time point, the boundaries of the metabolic model could be dynamically adjusted over the course of the day (Fig. 4).

**Fitting discrete transcriptomic data with continuous functions.** To build this data-driven model of *Chlamydomonas reinhardtii*, we used published transcriptomic data from cells grown in 12-h:12-h day-night cycles in a mixed photobioreactor with ambient air bubbling, replete nitrogen, and 200 $\mu$E of light (11). This data set provided 2-h resolution (or less) of changes in gene expression across the entire genome of *Chlamydomonas*. To use these data as constraints for a model with smaller time steps, we first needed to fit the discrete data points with continuous functions. After a quick visual inspection of the data, we chose 8 different types of transcript expression models that would fit the different expression profiles well: one-term cosine model, two-term cosine model, hat model, fixed-decay model, variable-decay model, constant model, Kronecker delta model, and cosine multiplicative model (see Fig. S1). For every gene in the transcriptome, we first estimated the parameters for each of the 8 proposed expression profiles which provided the best fit (see Fig. S1). Then, we used the corrected Akaike information criterion (AICc) (55, 56) to identify the profile that provided the best fit with the least loss of information.

The parameter estimation for each transcript expression model was performed with scipy.optimize.curve_fit. As there was no guarantee of global optima, the initial conditions for the fitting protocol were randomly perturbed and refit 100 times, with the lowest value for the AICc fit being selected for that particular model. Then, all the transcript expression models were compared against each other on the basis of AICc; the model with the lowest overall AICc was assumed to be the best model for this specific transcript. This protocol was repeated for every single transcript, thereby transforming thousands of sets

**FIG 3** Legend (Continued)

outperforming the *sta6* mutant (overall biomass in black line). Without acetate (only $CO_2$), the model predicts much slower growth for the WT strain and no growth for the *sta6* strain. (B) Predicted lipid content for WT and *sta6* strains in constant light (solid bars) and in diurnal light (lined bars). (C) Phenotype of individual gene knockouts for every gene in the metabolic model, once again using unitless normalized mass. Each vertical line in the graph represents a single gene knockout; the two indicator lines (STA6 and Wild Type) show the locations of each of those strains. The parent strain (WT) doubles once in 24 h, and thus, its mass ratio is 2. The majority of gene knockouts have the same final biomass as the WT, but there are many that result in severe defects in growth (on left side of the graph) and several that result in higher biomass. Chlb, chlorophyll *b*; Chla, chlorophyll *a*.

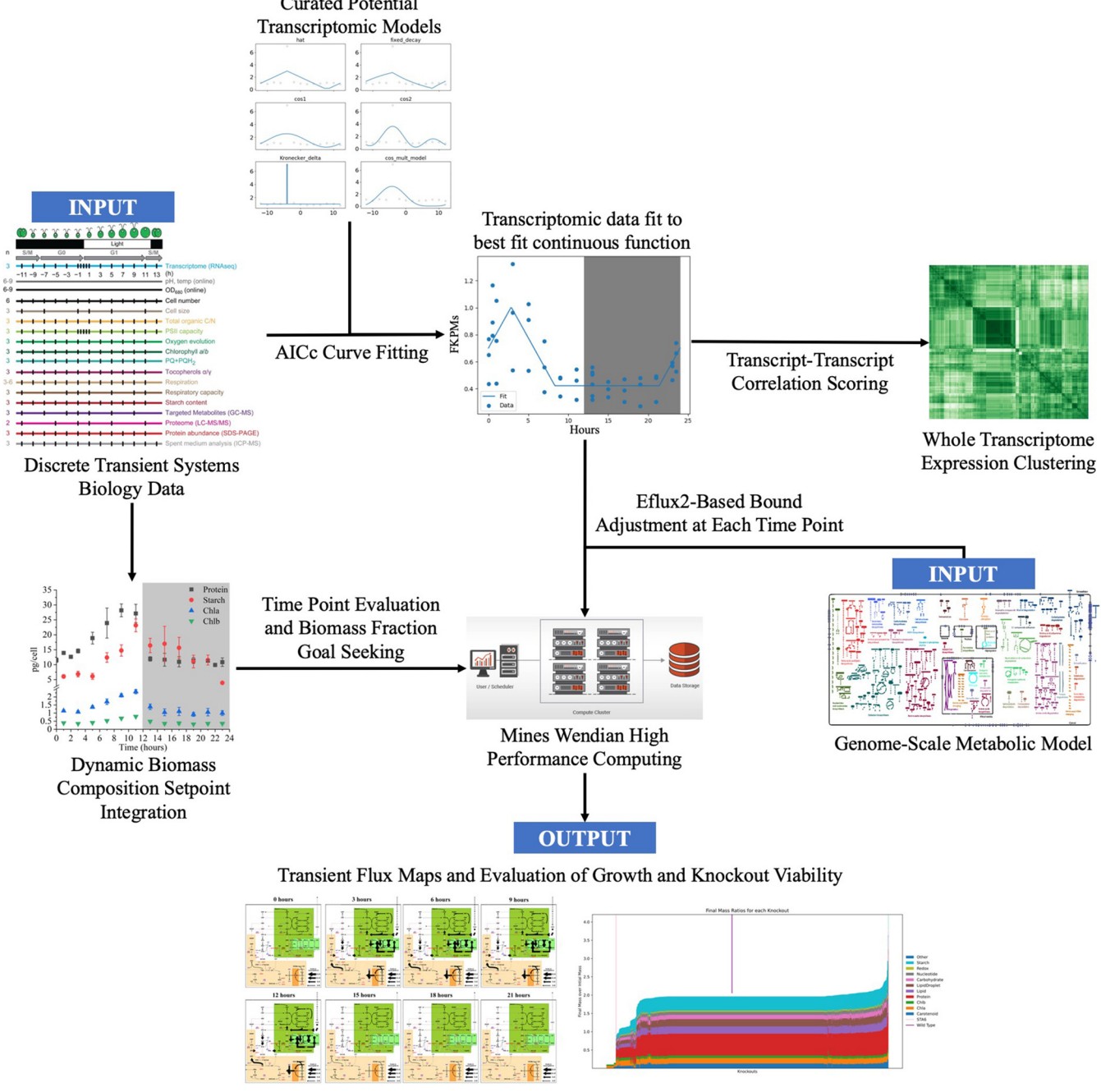

**FIG 4** Data pipeline used to create a dynamic metabolic model and a set of correlations. This can be utilized for any algal cell that has these data sets. Transcriptomic correlations aid in the discovery of new interactions between genes; the mechanistic metabolic model enables testing of knockouts and medium adjustments under dynamic diurnal conditions. The process takes the form of a flowchart; one information stream flows from "Discrete Transient Systems Biology Data" into "Dynamic Biomass Setpoint Integration" and from there into "Mines Wendian High Performance Computing" via "Time Point Evaluation and Biomass Fraction Goal Seeking." Another flows from "Curated Potential Transcriptomic Models" to "Transcriptomic Data Fit to Best Fit Continuous Function" and from there to both the endpoint of "Whole Transcriptome Expression Clustering" via "Transcript-Transcript Correlation Scoring" and to "Mines Wendian High Performance Computing," drawing from "Genome Scale Metabolic Model" along the way via "Eflux2-Based Bound Adjustment at Each Time Point." "Mines Wendian High Performance Computing" flows into "Transient Flux Maps and Evaluation of Growth and Knockout Viability," the final endpoint.

of discrete data points into a collection of curves that represented the cell's circadian rhythms.

**Clustering genes by expression profile.** The curves assigned to specific transcripts can also be used to evaluate the similarity between each transcript. This is typically performed in two ways: score and integral similarity. As described below, score is based on vectorizations of the parameters for each curve: by calculating the angle and distance between two fits, a quick measurement of curve similarity can be assessed. Integral similarity between curves is found by sampling the best-fit curve for each

transcript at small intervals, linearly mapping the sampled values such that the minimum sampled value is zero and the maximum sampled value is one in order to normalize each, and finally comparing the differences between the normalized curves at each sample point. Those with a high integral of difference are very dissimilar curves, while those with a very small integral have highly similar expression shapes (though they may have large differences in expression magnitudes). By calculating these measurements for every possible pair of transcriptomic interactions and clustering the results, we generate a heat map that summarizes the relationships between the circadian expressions of every gene. This map can be built for either method of comparison, either score or integral; the clustering in this study was performed with the integral method.

**Scoring.** The score method is faster than the integral method at examining correlations between transcripts and provides a good "first look" at potential clusters. However, it is also much more abstracted than the integral method. By vectorizing the parameters of the curves associated with each transcript, we can compare two vectors with relatively few calculations. Minimizing calculations is critical, as the number of comparisons scales as $O(n^2)$.

In order to vectorize a given expression curve, we begin by calculating the weight of the transcript expression models associated with the curve. A model's weight is calculated by comparing the AICc to the AICcs of all models summed together in the following fashion, where $\Delta_i$ is the difference between the AICc of model $i$ and the lowest AICc within the set of transcript expression models associated with the curve (here represented by $R$):

$$w_i = \frac{\exp\left(-\frac{1}{2}\Delta_i\right)}{\sum_{r=1}^{R}\exp\left(-\frac{1}{2}\Delta_r\right)} \qquad (1)$$

Weight is roughly conceptually equivalent to the probability that a specific model is the best of all given models at fitting the data. Repeating the above equation for all models within the set $R$ shows that the weights of all model fits for a particular set of data must sum to 1 (57). When we multiply the weight of a given transcript expression model by the parameter values that provide the best fit for that model, we transform the fit into a vector that represents both the fit's parameters and the utility of said fit; repeating this calculation for every model and concatenating all the resulting vectors into one gives us a vector that encapsulates all of the transcript expression models simultaneously.

These vectors enable inexpensive comparison between different transcripts. Because vector operations are, computationally speaking, relatively inexpensive, we can quickly run a number of calculations for each vector comparison. We first calculate the distance between each pair of vectors, using the Euclidian norm. The distance between a pair of vectors can indicate the similarity of fit parameters; even if two curves heavily favor the same model, the difference in parameters will result in a large distance. However, two curves that weight different models strongly may still have a small Euclidian distance between them. To account for this, we also calculate the angle between the vectors. If one transcript heavily weights a particular model while another places a significant weight on another, it is likely that the shape of the transcript expression is quite different. When the angle is calculated between them, it will reflect that information and approach orthogonality. We then combine these pieces of information in a multiplicative fashion, so that if either distance or angle indicates dissimilarity, the score will reflect that fact. While the cosine of the angle is already conveniently normalized between $-1$ and 1, it is desirable to also map the score onto a similar space. To do so, we generated a distance value:

$$\min\left(1 - \frac{\log(d)}{\log(d_m)}, 1\right) \qquad (2)$$

where $d$ is the distance between the vectors and $d_m$ is the maximum distance for all gene identifier (ID) vector pairs. This is neatly locked between 0, when the distance of the pair is the maximum distance in the entire data set, and 1, when the distance is so close as to be negligible. Moreover, the log operations shift the sensitivity of the curve so that most of the range of distance values from 0 to 1 are mapped onto relatively small distance values, thereby highlight small differences in distance even when maximum distance values are large. Multiplying the cosine of the angle by the distance value gives the score expression:

$$S = \cos\theta * \min\left(1 - \frac{\log(d)}{\log(d_m)}, 1\right) \qquad (3)$$

It is immediately clear that the only way to approach the maximum score of 1 is to have vectors that are closely aligned and of similar magnitudes. Should two vectors not be closely aligned, they cannot approach 1, regardless of how distant they are from each other—and should two vectors have near-identical direction but different magnitudes, their score will also be low.

**Image construction.** For every clustering map, the data frame containing the relevant data (either score or integral values for each pair) is loaded. As this file is generally around 6 GB, it is infeasible to generate the entire map at once. Instead, a set of linkages is created using scipy's cluster.hierarchy. linkage with the ward map. The linkages are then used to produce a truncated dendrogram with a set number of leaves. Each leaf is then processed into a nonsquare heat map, where every pixel represents

one pairwise comparison between transcripts. The maps are made iteratively, with each being cleared out of the memory before the next is added. Once all the maps are complete, the data frame is no longer required and can be cleared out as well, leaving space for each map to be loaded back in and stitched together into a final square map. For the weight sidebar, each model weight is included as a section of a stacked bar graph, which is saved separately and then added to the square map. Additional information can be added if required, such as the highlighting of specific genes.

**Using transcriptomic data to constrain solution space with E-Flux2.** Simulating a cell with a constraints-based approach requires constraints, which ideally reflect actual physiological or environmental phenomena. Transcripts can be integrated into these models in order to inform these constraints, using approaches like the E-Flux2 methodology (58); this method multiplies the level of transcriptomic expression, normalized to a fraction of the maximum, by the maximum allowable flux for all reactions and sets the result as the maximum allowable flux for that specific transcript. In this simulation, the maximum allowable flux through any reaction was set by matching the maximum light uptake through the transcriptomic constraints to the maximum amount of light in the environment.

We then extended this method by dynamically shifting the constraints of all reactions with transcriptomic linkages at every time point. At each time point, each curve was interrogated and normalized against the maximum for that curve over the day, and the bounds were lowered to that normalized fraction. When a reaction is catalyzed by a multimeric enzyme requiring the simultaneous expression of multiple genes, then the bound for said reaction is calculated by setting it to the minimum expression level of all the required genes, defined by the Boolean logic AND. Conversely, some reactions can be catalyzed by multiple enzymes; in this case, the reaction bound is calculated by proportionally setting it equal to the sum of transcript abundance of all associated gene products (defined by the Boolean OR). This is depicted mathematically below; equation 4 defines the normalization function, equations 5 and 6 are the equations for AND and OR relationships, respectively, and equations 7, 8, and 9 show an example.

$$v_a^t = \frac{a(t)}{\max\{a(t) : t = (0, 24)\}} \tag{4}$$

Equation 4's resulting value, $v_a^t$, represents the normalized flux fraction for a given gene $a$ at time $t$, where $a(t)$ represents the value of the best-fit curve for gene $a$ at time $t$.

$$v_\wedge = \min(v_a^t, v_b^t) \tag{5}$$

$$v_\vee = v_a^t + v_b^t \tag{6}$$

$$R_A = a \vee (b \wedge c) \tag{7}$$

$$v_a(t) = v_a^t + \min(v_b^t, v_c^t) \tag{8}$$

Equation 7 is a mathematical representation of the gene-reaction rule $a$ OR ($b$ AND $c$); equation 8 is the corresponding representation of how to calculate the flux fraction for the same rule.

$$u_A(t) = u_{max} \times \min[1, v_a(t)] \tag{9}$$

Finally, the flux fraction is multiplied by the maximum reaction flux of the system to calculate the bound for the reaction $A$ at time $t$. This value is used for the upper bound of the reaction. If the reaction is reversible, the negative of it is used for the lower as well. Otherwise, the lower bound of the reaction is zero. The maximum reaction flux of the system was calculated such that the maximum rate of light uptake over the course of the day matched the light availability of the system. This approach is feasible, despite the discrete nature of the transcriptomic data set, because the process of fitting the curves to the point creates a continuous representation of the cell's transcriptome that can be queried at any arbitrary time. It is worth noting that the confidence of this approach is a function of the collection resolution; faster and more transient events demand smaller intervals between data points, as implied by the Nyquist-Shannon sampling theorem.

**Decoupling biomass and transient objectives.** A typical constraint-based model formulation uses a static biomass formation equation to represent metabolites being used for the creation of biomass. Such a fixed equation is inappropriate for a circadian model, both because the composition of the cell changes over the diel cycle and because the cell's capability to make individual components of the broader equation may change over time. In order to accommodate this difference, we split the cell's biomass equation into subgroups and laid a control system over the top of it, such that the model would always attempt to aim for a long-term stable solution that approached the known biomass composition at different times of the day. The decoupled biomass equation consists of 10 components: carotenoid, chlorophyll *a*, chlorophyll *b*, protein, lipid, lipid droplet, carbohydrate, nucleotide, redox, and starch; details of these are in Data Set S1. This division of biomass allows the model to produce different components of biomass when it can, a freedom of production that is a realistic representation of diurnal growth because it enables the model to replicate the shifting biomass composition that synchronized cells experience over the course of the day. Additionally, the split enables the cell to allocate energy and carbon even if a specific biomass metabolite cannot be made, a critical modification. If this were not the

case, a transcriptomic limit or other constraint that prevented even one biomass metabolite from being made at any specific time point would compromise the cell's ability to accumulate any energy at all during that time point. With this approach, only the biomass component directly associated with that metabolite cannot be produced; the cell simply produces other biomass components during that specific time point instead, and then, once the model iterates to a time point that makes production of that missing biomass component once again feasible, the control system that manages the objective function of the model weights the production of that missing biomass component to make up for the shortfall. In this way, the model can experience transcriptomic limits and still hew realistically close to the biomass composition of the cell, without having overall biomass production be compromised by transient inabilities to produce individual metabolites.

In order to properly treat lipid production, we separated the lipid components of biomass into both a general lipid component and a lipid droplet component. The lipid droplet contained all triacylglycerides in the original biomass equation that had tails measured in the work of Wang et al., while the general lipid class contained all other lipids (59). This separation of lipids allows the lipid droplet fraction to rise and fall independently of the other lipid components. This behavior is pertinent for modeling lipid fraction under nitrogen stress, as the general lipid component contains lipids that contain nitrogen and therefore will be constrained in times of low nitrogen availability.

**Setting catabolism and anabolism rates.** After the model is constrained to a time point, the production possibilities need to be evaluated. Critically, this means that the production space for biomass components must be defined. In this modeling approach, the biomass equation of the original FBA model—iCre1355 (39) in this implementation, though this approach is not limited to only this model—is split into components, so that not all biologically relevant metabolites need to be available in the correct ratios for any of them to be consumed. Additional details of the model split are available in Text S1 in the supplemental material. For every time point in the diel cycle, the production of each component is tested: first for production from just the exchange reactions available and then for production from consumption of each of the other biomass components.

Once the time points of the daily cycle have been evaluated, the model begins to run. At each time point, the model is constrained according to the transcriptomic curves, and the production goals and consumption preferences are set by the error terms for the components. These error terms are calculated using a pair of proportional integral (PI) control systems—one for the production terms and one for the consumption terms—thereby abstracting the complex cellular regulatory networks that govern anabolic and catabolic processes into a simpler parameter space. After the production and consumption error terms for a given time point have been calculated, the model tries to produce the non-growth-associated maintenance ATP (NGAM) with the medium; if it cannot, it will then consume one or more components of biomass to stay alive.

Component consumption is prioritized by the consumption error terms; those with the highest positive error (i.e., the component in question is likely above its setpoint and has been for some time) are consumed first. Once the cell has made NGAM, it evaluates production potential. If the medium can be successfully consumed (i.e., it can utilize light to produce NGAM), or if any components with positive degradation error remain after making ATP, the cell will attempt to produce components from either the medium or the positive-error components, respectively. The production possibilities are precalculated for each time point and possible energy source, either medium or specific components. In the event that metabolites in a component cannot be catabolized by the cell, they are assumed to be dumped out of the cell entirely, resulting in an overall mass and energy loss. The cell uses the production error terms and the production possibilities to calculate which components are produced at what ratios. These components and ratios form the initial objective function for the cell's simulation step.

During the production phase, the cell attempts to produce as much of the initial objective as possible. Once it reaches a limitation, the model fixes the production flux, then checks which component is limiting production, and attempts to make more of the nonlimiting desired ones. This iterative expansion of the production window continues until there are no more desired components or—more commonly—the constraints of the cell limit overall production. Once components have been consumed and produced, the rates of production and consumption are multiplied by the total cell mass to generate the rates at which each biomass component is changing. This rate represents how much of each component has accumulated or been consumed since the previous timestep. Multiplying it by the timestep yields the change in biomass components across the length of the timestep, thus linking the cell's metabolism to its overall biomass. By repeating this entire process each and every timestep, including the readjustment of transcript bounds and the recalculation of new objective functions based on how the new biomass compositions differ from the existing setpoints, the model can effectively keep track of biomass amounts, accommodate dynamic constraints that make individual reactions infeasible at certain times and prioritize the production or degradation of biomass components in ways that recapitulate energetic and experimental influences. In the current approach, the model simulates growth in 6-min timesteps, so the timesteps occur 10 times every simulated hour. Due to the continuous nature of the transcriptomic fitting process, that timestep is arbitrarily adjustable if more or less resolution is required for different applications.

**Universal single-knockout evaluation.** In order to demonstrate the utility of this modeling approach, we conducted a survey of all gene knockouts within the model. Using high-performance computing (HPC), we were able to simultaneously evaluate large numbers of genes. To simulate each gene knockout, the transcriptomic fits for the gene in question were set to zero, thereby simulating a complete knockout. The knockout mutants were simulated for 168 h, a full week, at light and non-growth-associated maintenance ATP light levels that produced approximately one doubling in the wild-type cell over the length of the simulation. While these conditions differ from other simulations, they were selected for specific reasons: the long growing period gives the cells time to acclimate to the mutants and stabilize their own control loops, while the

reduced available energy means that impacts of the mutations can more clearly be seen. For every run, the full biomass composition was recorded at the end of the simulation, and the impacts of each gene on the overall growth and biomass composition can be clearly seen.

**Data availability.** Transcriptomic data were drawn from reference 11. Additional result files and source code are available at https://github.com/metcalex/Transcriptomic_Circadian_Modeling_Supplemental.

## SUPPLEMENTAL MATERIAL

Supplemental material is available online only.

**DATA SET S1**, XLSX file, 0.2 MB.

**MOVIE S1**, MOV file, 2 MB.

**TEXT S1**, DOCX file, 0.01 MB.

**TEXT S2**, DOCX file, 0.01 MB.

**FIG S1**, TIF file, 2 MB.

## ACKNOWLEDGMENTS

We thank Sabeeha Merchant and Daniela Strenkert for their valuable discussion during the development of the model.

This research was supported by the DOE Office of Science, Office of Biological and Environmental Research (BER), grant no. DE-SC0019171.

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
