## [Reviewer comments · mSystems]

Rhythm of The Night (and Day): Predictive metabolic modeling of diurnal growth in *Chlamydomonas*

Alexander Metcalf and Nanette Boyle

Corresponding Author(s): Nanette Boyle, Colorado School of Mines

Review Timeline:

Submission Date:	February 25, 2022
Editorial Decision:	March 22, 2022
Revision Received:	April 29, 2022
Accepted:	May 24, 2022

Editor: Hans Bernstein

Reviewer(s): Disclosure of reviewer identity is with reference to reviewer comments included in decision letter(s). The following individuals involved in review of your submission have agreed to reveal their identity: Daniela Morales-Sanchez (Reviewer #2)

Transaction Report:

DOI: <https://doi.org/10.1128/msystems.00176-22>

March 22, 2022

Prof. Nanette R Boyle
Colorado School of Mines
Chemical & Biological Engineering
1613 Illinois St
Golden, CO 80401

Re: mSystems00176-22 (**Rhythm of The Night (and Day): Predictive metabolic modeling of circadian growth in *Chlamydomonas***)

Dear Prof. Nanette R Boyle:

Thank you for submitting your manuscript to mSystems. We have completed our review and I am pleased to inform you that, in principle, we expect to accept it for publication in mSystems. However, acceptance will not be final until you have adequately addressed the reviewer comments.

Please respond to each of reviewer #1's comments rigorously by making changes to the manuscript where possible. Reviewer #2 made a comment about the quality of the figures. I think the figure quality is in fact very good, but please please look critically at the text size and readability in Figure 4.

Also there are more than 10 supplementary documents. Please reduce these to the minimum possible number, which to me looks like three (movie, Document S1 and combined sheets).

Preparing Revision Guidelines

Sincerely,

Hans Bernstein

Editor, mSystems

Journals Department
Reviewer comments:

Reviewer #1 (Comments for the Author):

Comments and suggestions are included in the attached file.

Reviewer #2 (Comments for the Author):

Dear authors, these are my comments:

1. I suggest checking the grammar. I am not an English native speaker but I found some mistakes in the grammar.
2. Which gene(s) did you use as housekeeping gene for transcript abundance normalization then? Did you check the actin gene? Is the most used for normalization in Chlamy.
3. The quality of the figures is very poor or at least it's what I see in the pdf. I suggest improving them.
4. Figure 2 and the animated figure are too messy. It's there any other better way to show the metabolic fluxes? It does not look good with these thick arrows.
5. This model is for Chlamy, a model algae, but would it work in other commercial algae?
6. Figure 3A and C, overall mass axis needs units?
7. Figure 3 legend, ..."In the presence of acetate, both strains grow well, " what do you mean with grow well? Maybe a more accurate word here?
8. Figure 4 legend. Would be good to have a description of the figure here.
9. Did you perform any statistical analysis to validate your results?

Review for article mSystems00176-22 “Rhythm of the Night (and Day): Predictive metabolic modeling of circadian growth in *Chlamydomonas*”

Summary:

This article presents an implementation of time-resolved transcriptomic data with flux balance analysis (FBA) to produce a dynamic version of expression-based stoichiometric metabolic modeling. Using discrete measurements of transcripts from another study, the authors developed continuous functions estimating transcript level (testing several different function estimations, assessing and determining the best fit for each gene). They implemented these transcript expression functions as additional constraints in the FBA framework to iteratively adjust the bounds for each metabolic reaction according to expression at each time step, also incorporating the capacity for changing biomass composition at each time step. The authors show that they can model the growth of wild type *Chlamydomonas reinhardtii* under diel conditions using the given data, and they also investigate the predicted effects of different gene knockouts, focusing mainly on predictions of the *sta6* mutant which is unable to accumulate starch.

While the predictions using the gene expression-based dynamic FBA and the resulting flux map animations are certainly impressive, I think the impact and application of the article can be improved through the modifications suggested below, and the connection to the sustainable energy field can be strengthened.

Comments:

1. Alternate phrasing should be considered for the title in place of “circadian growth” – either “circadian rhythm” or “diurnal growth”, by convention.
2. The introduction focuses on the impact of light-dark cycles on economical growth of algae for sustainable fuels. However, the predictions show that WT *C. reinhardtii* is not expected to accumulate lipid droplets under nitrogen limitation in diurnal light as they are observed to do under constant light, which seems to undercut the rationale for pursuing the work presented in the introduction. The discussion and conclusion should more fully address this point and expand upon the potential applications.
3. It would be helpful for the introduction to include a bit more information on what kinds of algal production strains have been developed to date, to more clearly emphasize how the presented work can expand upon what has already been done.
4. The previous study from which the transcriptomic data was obtained is cited (Strenkert et al., 2019), but the experimental scenario should be explained further beyond the 12:12 light:dark criterion to better understand the conditions that the gene expression data represents (e.g., light intensity, CO₂ availability, nitrogen source, etc.).
5. It is noted in line 115 that the sudden onset of light in the experimental conditions does not accurately mimic a natural environment – a more in-depth exposition of how this factor might affect the model results and the metabolic patterns interpreted from the

model results would be helpful. Would simulation predictions be expected to be quite different?

6. The role of photoinhibition / photorespiratory pathways is not mentioned, yet these pathways are likely very important as high light intensities are commonly encountered during mid-day in natural environments. A discussion of the impact of photorespiration on productivity would be helpful.
7. A few details of the model implementation remain unclear: what time scale resolution is used in the dynamic FBA simulations? How is the changing objective function decided at each iteration (the initial objective is explained in lines 479-484, but the process for updating the biomass composition upon each iterative cycle is vague)? How realistic is it to use a changing biomass equation if it allows growth without requiring all biomass metabolites (line 437-439)?
8. The gene-reaction rules AND and OR described in lines 402-405 seem to be defined backwards. AND would imply that both genes are used, which would involve a sum of the fluxes; OR would imply one or the other reaction, which would involve a minimum of the two fluxes. Additional explanation of this logic would help clarify.
9. The knockout evaluation methods described in lines 495-497 state that a 168-hour simulation assumed a single doubling over that time frame, which seems extremely slow and conflicts with the information given in the Figure 3 legend which gives a doubling time of 24 hours; this apparent inconsistency should be clarified.
10. The arrangement of Figure 3C makes it confusing to understand the role of the two lines for *sta6* and WT on the graph. It seems strange to show the WT when the x-axis represents knockouts, and they appear to have a final / initial mass value of 4, which is not possible for the WT according to the figure caption (reports a final / initial mass value of 2).
11. The modeling of mutant phenotypes presents some results that could be predicted as possible which are not actually feasible and would require experimental testing, due to the variable biomass composition and the way the biomass components are defined. A few examples of detecting false positives are listed, but what percent of false positives might be expected with this method? It represents a potentially quite cumbersome process to evaluate and check results, so addressing the impact of false positives would be important for assessing how useful mutant simulations are. Could it be a potential solution to employ a standard formulation FBA model and simply run two separate simulations (one with light and one without light) to test the feasibility and avoid some of these false positives?
12. The conclusion / discussion should better address how the presented method can be extended or applied to other organisms. Currently the presented work serves as a very nice case study with a specific data set, but translation to other organisms and clarifying the utility for bioengineering applications as stated in the introduction is important to improve the impact. For example, what is the time scale resolution of discrete transcriptomics measurements that is needed to accurately model the transcript

changes? The data set used was two hours apart, but many experiments are not able to use such a fine level of resolution – how much molecular data is needed to be useful?

13. Several typographical and grammatical errors throughout the document still require attention: e.g., “outcomes [of] a starchless mutant” (line 22), spell out *Arabidopsis* (line 74), avoid use of contractions like “don’t” (line 77), run-on sentences (lines 77-78, 112), “constraint-based” rather than “constraints-based” (line 88 and others), “dynamic model’s” rather than “dynamic’s model” (line 187), “Phtyozome” (line 239), “Kronkecker delta” (line 294), “vectorize” rather than “vectorized” (line 331), among others.

Response to Reviewers for mSystems00176-22 “Rhythm of the Night (and Day): Predictive metabolic the modeling of circadian growth in *Chlamydomonas*”

Below is our point by point response to reviewers. Our response is shown in red text. Text copied over from the manuscript for ease of review is shown in italics, while line numbers from the marked-up manuscript pdf are included for ease of searching.

Notes from editor:

- I think the figure quality is in fact very good, but please please look critically at the text size and readability in Figure 4.

We have reexported Figure 4, but we should be clear that the smaller figures in the main figure are more for aesthetics than for interpretation so we weren't necessarily concerned with readability. If the editor would like us to remake this figure with larger text in the subfigures, we are happy to.

- Also there are more than 10 supplementary documents. Please reduce these to the minimum possible number, which to me looks like three (movie, Document S1 and combined sheets).

We have condensed the excel files into one file so now there are only 3 supplemental files: the word document, the excel file and the movie

REVIEWER 1:

Summary:

This article presents an implementation of time-resolved transcriptomic data with flux balance analysis (FBA) to produce a dynamic version of expression-based stoichiometric metabolic modeling. Using discrete measurements of transcripts from another study, the authors developed continuous functions estimating transcript level (testing several different function estimations, assessing and determining the best fit for each gene). They implemented these transcript expression functions as additional constraints in the FBA framework to iteratively adjust the bounds for each metabolic reaction according to expression at each time step, also incorporating the capacity for changing biomass composition at each time step. The authors show that they can model the growth of wild type *Chlamydomonas reinhardtii* under diel conditions using the given data, and they also investigate the predicted effects of different gene knockouts, focusing mainly on predictions of the *sta6* mutant which is unable to accumulate starch.

While the predictions using the gene expression-based dynamic FBA and the resulting flux map animations are certainly impressive, I think the impact and application of the article can be improved through the modifications suggested below, and the connection to the sustainable energy field can be strengthened.

Comments:

1. Alternate phrasing should be considered for the title in place of “circadian growth” – either “circadian rhythm” or “diurnal growth”, by convention.

Thank you for pointing this out. We have edited the entire document to either have circadian rhythm or diurnal growth (including the title).

2. The introduction focuses on the impact of light-dark cycles on economical growth of algae for sustainable fuels. However, the predictions show that WT *C. reinhardtii* is not expected to accumulate lipid droplets under nitrogen limitation in diurnal light as they are observed to do under constant light, which seems to undercut the rationale for pursuing the work presented in the introduction. The discussion and conclusion should more fully address this point and expand upon the potential applications.

Yes! This is an excellent example of why we need dynamic diurnal modeling because what has been tried in the past relies on physiological responses that have been characterized in continuous light, not diurnal light. Our model (as shown in the conclusion) is able to identify other approaches which may lead to the accumulation of lipids in algae in diurnal light. We have made this more clear in the introduction, see lines 71-82.

Currently, almost all metabolic engineering efforts in algae and cyanobacteria rely on growth in laboratory conditions with a continuous supply of light (9-22). This results in a steady state growth environment that more closely mimics that of heterotrophic bacteria and enables more straight forward design and engineering of cells. However, large scale growth of photosynthetic organisms necessitates growth in diurnal conditions outdoors, and the strong circadian rhythms that lead to dynamic gene expression can confound engineering efforts, as was reported by Cheah et al. (15). This also means that engineering strategies that have been shown to result in increased productivity in lab conditions will not directly translate to increased productivity in diurnal growth. Therefore, it is imperative that we develop tools that will enable more predictive and rational engineering of algal cells in diurnal growth.

3. It would be helpful for the introduction to include a bit more information on what kinds of algal production strains have been developed to date, to more clearly emphasize how the presented work can expand upon what has already been done.

The introduction already has a number of citations for metabolic engineering efforts in photosynthetic organisms (see lines 73, 77, 80). As far as we know, an engineered strain of algae has not yet been deployed for large scale growth outdoors; most industrial production of lipids or nutraceuticals use wild type strains. There are industrial strains of engineered algae that are grown heterotrophically for production of designer lipids, but that is not the intended use of our model since heterotrophic growth is steady state.

We have added an additional line in the introduction to note this (see lines 77-82)

This also means that engineering strategies that have been shown to result in increased productivity in lab conditions will not directly translate to increased

productivity in diurnal growth and is one reason why most current industrial algal production uses wild type strains (23).

4. The previous study from which the transcriptomic data was obtained is cited (Strenkert et al., 2019), but the experimental scenario should be explained further beyond the 12:12 light:dark criterion to better understand the conditions that the gene expression data represents (e.g., light intensity, CO₂ availability, nitrogen source, etc.).

Thank you for the feedback; we've added some experimental information in lines 502-505, reproduced below:

To build this data-driven model of Chlamydomonas reinhardtii, we used published transcriptomic data from cells grown in 12:12 day night cycles in a mixed photobioreactor with ambient air bubbling, replete nitrogen, and 200 μE light (1).

Should the reader want more info about the experimental conditions, they can read the referenced paper.

5. It is noted in line 115 that the sudden onset of light in the experimental conditions does not accurately mimic a natural environment – a more in-depth exposition of how this factor might affect the model results and the metabolic patterns interpreted from model results would be helpful. Would simulation predictions be expected to be quite different?

Thank you for raising this point; we've addressed it in the document in lines 188-206, reproduced below for clarity:

This ramping supports that the instantaneous light switch likely has a somewhat minimal overall effect on the cell's phenotype and growth relative to the gradual rise and fall in illumination the cell would experience under natural light. While some stress transcripts are produced at the moment of illumination, the transcript ramps associated with energy production still combine to produce light uptake bounds that rise and fall over the course of the day, reflecting the natural ramp of outdoor light. This consistency implies that the immediate light switch was not particularly impactful on energy availability – if it were, we would expect to see that light harvesting transcripts become immediately expressed at full force and persist for the length of the day. As light uptake is the limiting overall factor for autotrophic growth in non-nutrient limited conditions, the rise and fall implies that the cell's growth is not drastically affected by the sudden onset of light. However, it is possible that the internal constraints on the cell's metabolism are shifted by the onset; that is, the cell still experiences roughly the same ramp up and down of light harvesting that it would under natural light onset, but - due to changes in the expression patterns of other transcripts associated with other subsets of metabolism – the cell allocates the energy captured from the light it receives differently. This change would result in the cell producing different components of biomass at different timepoints – but the overall effect on total biomass would likely be relatively minimal; the cell would just have a slightly different

composition at different timepoints. We therefore remain confident that the sudden light onset does not invalidate overall predictions of this model.

6. The role of photoinhibition / photorespiratory pathways is not mentioned, yet these pathways are likely very important as high light intensities are commonly encountered during mid-day in natural environments. A discussion of the impact of photorespiration on productivity would be helpful.

Photoinhibition is usually poorly captured in models, yes – we’ve added to the document lines 210-247 to clarify this, reproduced below:

Because constraints based modeling uses optimization to calculate flux vectors, if an inefficient reaction is not strictly required, the model will avoid allocating flux through it. This phenomena can also be seen in the model’s lack of photorespiration and photoinhibition. As these processes consume energy without increasing biomass and are not enforced by constraints, the model will invariably allocate no flux to them. Verifying the magnitudes of such inefficiencies and adding in modeling constraints to account for them are areas of potential future investigation that could increase the accuracy of the model under certain conditions where such processes may predominate. Two potential conditions of interest on this front are high light or low carbon availability, as these conditions encourage photoinhibition and photorespiration respectively – but because these conditions are not encountered in either the model’s source data set or the simulated environments, we believe the model still captures a significant portion of the cell’s metabolic processes.

7. A few details of the model implementation remain unclear: what time scale resolution is used in the dynamic FBA simulations? How is the changing objective function decided at each iteration (the initial objective is explained in lines 479-484, but the process for updating the biomass composition upon each iterative cycle is vague)? How realistic is it to use a changing biomass equation if it allows growth without requiring all biomass metabolites (line 437-439)?

We are sorry that this wasn’t more clear in the paper. We have added additional details to the manuscript in order to make this more clear. Details of time scale resolution and updating have been added in lines 757-771 in the revised manuscript, reproduced below, while further explanation of the changing biomass growth has been added in lines 689-704.

Once components have been consumed and produced, the rates of production and consumption are multiplied by the total cell mass to generate the rates at which each biomass component is changing. This rate represents how much of each component has accumulated or been consumed since the previous timestep. Multiplying it by the timestep yields the change in biomass components across the length of the timestep, thusly linking the cell’s metabolism to its overall biomass. By repeating this entire process each and every timestep, including the re-

adjustment of transcript bounds and the re-calculation of new objective functions based on how the new biomass compositions differ from the existing setpoints, the model can effectively keep track of biomass amounts, accommodate for dynamic constraints that make individual reactions infeasible at certain times, and prioritize the production or degradation of biomass components in ways that recapitulate energetic and experimental influences. In the current approach, the model simulates growth in six minute timesteps, so the timesteps occurs ten times every simulated hour. Due to the continuous nature of the transcriptomic fitting process, that timestep is arbitrarily adjustable if more or less resolution is required for different applications.

This division of biomass allows the model to produce different components of biomass when it can, a freedom of production that is a realistic representation of diurnal growth because it enables the model to replicate the shifting biomass composition that synchronized cells experience over the course of the day. Additionally, the split enables the cell to allocate energy and carbon even if a specific biomass metabolite cannot be made, a critical modification. If this were not the case, a transcriptomic limit or other constraint that prevented even one biomass metabolite from being made at any specific timepoint would compromise the cell's ability to accumulate any energy at all during that timepoint. With this approach, only the biomass component directly associated with that metabolite cannot be produced; the cell simply produces other biomass components during that specific timepoint instead – and then, once the model iterates to a timepoint that makes production of that missing biomass component once again feasible, the control system that manages the objective function of the model weights the production of that missing biomass component to make up for the shortfall. In this way, the model can experience transcriptomic limits and still hew realistically close to the biomass composition of the cell, without having overall biomass production be compromised by transient inability to produce individual metabolites.

8. The gene-reaction rules AND and OR described in lines 637-642 seem to be defined backwards. AND would imply that both genes are used, which would involve a sum of the fluxes; OR would imply one or the other reaction, which would involve a minimum of the two fluxes. Additional explanation of this logic would help clarify.

We apologize if this point wasn't clear. The gene-protein-reaction rules operate off Boolean logic applied to the transcriptomic data, not the reactions. AND means that both gene products are required to make the enzyme needed to catalyze the reaction(s), and therefore, the bound is set by the minimum of either gene. OR means that either gene product is sufficient to produce the enzyme to catalyze the reaction, therefore the bound is set by the sum of both genes. This relationship has clarified in the manuscript in lines 563-568 (reproduced below), and is also elaborated in the article that is cited in the section in question.

When a reaction is catalyzed by a multimeric enzyme requiring the simultaneous expression of multiple genes, then the bound for said reaction is calculated by setting it to the minimum expression level of all the required genes, defined by the Boolean logic AND. Conversely, some reactions can be catalyzed by multiple enzymes; in this case, the reaction bound is calculated by proportionally setting it equal to the sum of transcript abundance of all associated gene products (defined by the Boolean OR).

9. The knockout evaluation methods described in lines 495-497 state that a 168-hour simulation assumed a single doubling over that time frame, which seems extremely slow and conflicts with the information given in the Figure 3 legend which gives a doubling time of 24 hours; this apparent inconsistency should be clarified.

Thank you for mentioning a possible point of clarification here. Light availability is a changeable constraint in this simulation, so we lowered the light availability to place more stress upon the cells and reduce the energy budget, thus making it more clear which cells might be suffering from significantly detrimental mutations. We added a similar note in the manuscript, lines 777-785, which are reproduced below:

The knockout mutants were simulated for 168 hours, a full week, at light and non-growth-associated maintenance ATP light levels that produced approximately one doubling in the wild type cell over the length of the simulation. While these conditions differ from other simulations, they were selected for specific reasons: the long growing period gives the cells time to acclimate to the mutants and stabilize their own control loops, while the reduced available energy means that impacts of the mutations can more clearly be seen.

10. The arrangement of Figure 3C makes it confusing to understand the role of the two lines for sta6 and WT on the graph. It seems strange to show the WT when the x-axis represents knockouts, and they appear to have a final / initial mass value of 4, which is not possible for the WT according to the figure caption (reports a final / initial mass value of 2).

We appreciate the opportunity to provide some graphical clarification. The lines point to where the mutants are located in the graph. WT is included as a strain for the purposes of comparison. Clarified in manuscript, line 987:

the two indicator lines (STA6 and Wild Type) show the locations of each of those strains

11. The modeling of mutant phenotypes presents some results that could be predicted as possible which are not actually feasible and would require experimental testing, due to the variable biomass composition and the way the biomass components are defined. A few examples of detecting false positives are listed, but what percent of false positives

might be expected with this method? It represents a potentially quite cumbersome process to evaluate and check results, so addressing the impact of false positives would be important for assessing how useful mutant simulations are. Could it be a potential solution to employ a standard formulation FBA model and simply run two separate simulations (one with light and one without light) to test the feasibility and avoid some of these false positives?

We apologize if this point wasn't clear. We do run the steady state, constant light FBA model with no transcriptomic constraints as an additional check, but it is explicitly noted in the manuscript that the sole utility of that process is checking for fatal mutations (because if a modeled mutation is fatal under steady state conditions with constant light, it will also be fatal under diurnal conditions.) Generally, these mutations are fatal because they fail to produce one or more biomass components; this is discussed extensively in the "Modeling Mutant Phenotypes" section, lines 261-274, which is reproduced below:

Due to the way FBA is formulated, it can only model steady state conditions, not the transient changes that occur during the switch back and forth. This can result in knockouts that are not fatal under steady state becoming so, such as Cre02.g145800.t1.2, a gene that encodes for mitochondrial malate dehydrogenase. As might be expected from a cell that heavily relies upon shuttling carbon to the mitochondrial ATP synthase, removing the cell's ability to do so efficiently results in long-term failure to thrive. There are also some knockouts that are predicted to overproduce in a circadian environment, despite being fatal in a steady state one – but these are universally the result of the dynamic model changing biomass constraints. Because the model does not strictly constrain the ratios of biomass components, the model can suggest that some knockouts are feasible despite their inability to produce biomass components; a good example of this is Cre17.g728950.t1.1, which controls production of flavin adenine dinucleotide (FAD). This redox coenzyme is only required in small amounts in the biomass equation, but is large and therefore metabolically costly to produce. Mutants that cannot make it are predicted to grow faster, but this result is unlikely to be experimentally borne out.

12. The conclusion / discussion should better address how the presented method can be extended or applied to other organisms. Currently the presented work serves as a very nice case study with a specific data set, but translation to other organisms and clarifying the utility for bioengineering applications as stated in the introduction is important to improve the impact. For example, what is the time scale resolution of discrete transcriptomics measurements that is needed to accurately model the transcript changes? The data set used was two hours apart, but many experiments are not able to use such a fine level of resolution – how much molecular data is needed to be useful?

Thank you for your feedback on these topics. We've added more information in two locations; a note on broader applicability has been added to the conclusion

(lines 419-488, reproduced below), while a note on data resolution has been added to the methods in lines (lines 668-670):

When that predictive power is combined with the inherently parallelizable nature of in silico modeling, it is possible to quickly assess bioengineering approaches for feasibility, as demonstrated by individual simulations of a broad swath of single knockout mutants. Additionally, this approach requires relatively few discreet types of data –the primary requirements are only a pre-existing constraint based model, an appropriate transcriptomic data set, and a time-course of biomass. There is no other organism specific information required. Because of this, our approach can be generalized to any other photosynthetic algae species with similar data availability.

It is worth noting that the confidence of this approach is a function of the collection resolution; faster and more transient events demand smaller intervals between data points, as implied by the Nyquist-Shannon sampling theorem.

13. Several typographical and grammatical errors throughout the document still require attention: e.g., “outcomes [of] a starchless mutant” (line 22), spell out Arabidopsis (line 74), avoid use of contractions like “don’t” (line 77), run-on sentences (lines 77-78, 112), “constraint-based” rather than “constraints-based” (line 88 and others), “dynamic model’s” rather than “dynamic’s model” (line 187), “Phytozome” (line 239), “Kronkecker delta” (line 294), “vectorize” rather than “vectorized” (line 331), among others.

Fixed; thank you for spotting them!

REVIEWER 2

This manuscript reports the development of a dynamic metabolic model for diurnal growth of *Chlamydomonas reinhardtii* based on experimental data that predicts phenotype from genotype. It is very relevant because the implementation of this model, that includes the impact of genetic and environmental changes on the growth, biomass composition and intracellular fluxes, will allow faster design of processes for the production of high-value compounds and biofuels.

Comments to authors:

1. Check the grammar. I am not English native speaker but I found some mistakes in the grammar.

Thank you for noting this – we have re-read the document and corrected grammar mistakes.

2. Which gene(s) did you use as housekeeping gene for transcript abundance normalization then? Did you check the actin gene? Is the most used for normalization in Chlamy.

The transcriptomic data was obtained from the published paper:

D. Strenkert *et al.*, Multiomics resolution of molecular events during a day in the life of *Chlamydomonas*. *Proceedings of the National Academy of Sciences* **116**, 2374-2383 (2019).

As it is published in PNAS and the corresponding author (Sabeeha Merchant) is a world expert in the use of RNA-Seq for quantifying transcript abundance in *Chlamydomonas* (and other algae), we didn't feel the need to reprocess the data but took it as published.

We did actually look at the expression levels of the well-known *Chlamydomonas* housekeeping gene *RACK1* in the data (see Figure 1) and it extremely stable across the timecourse of experiments.

3. The quality of figures are very poor or at least it's what I see in the pdf.

We apologize for this – we believe the conversion of the PDF was bad so we have re-exported and for the final draft the figures will be uploaded as high resolution .tif files.

4. Figure 2 and animated figure are too messy. It's there any other better way to show the metabolic fluxes? It does not look good with these thick arrows.

Thank you for your feedback. It is very difficult to visualize metabolic pathways due to their complexity, and the added variable of time makes it even more complex. Visually speaking, we believe (and it is the convention for fluxes) it is easier to see changes in the metabolism when the arrow thickness changes.

5. This model is for Chlamy, a model algae, but would it work in other commercial algae?

Indeed! Assuming the data and model exists, it is absolutely a feasible conversion. The conclusion has been edited to reflect this in lines 419-488, reproduced below.

When that predictive power is combined with the inherently parallelizable nature of in silico modeling, it is possible to quickly assess bioengineering approaches for feasibility, as demonstrated by individual simulations of a broad swath of single knockout mutants. Additionally, this approach requires relatively few discreet types of data –the primary requirements are only a pre-existing constraint based model, an appropriate transcriptomic data set, and a time-course of biomass. There is no other organism specific information required. Because of this, our approach can be generalized to any other photosynthetic algae species with similar data availability.

6. Figure 3A and C, overall mass axis needs units?

Thank you for checking; these figures measure a normalized mass, and are therefore unitless. Edits have been made to reflect this in lines 979-986, reproduced below:

A) Growth and biomass composition simulations for WT and sta6 in diurnal light (the lighter sections of the plots represent the illuminated phase, while the darker sections represent darkness) with and without an organic carbon source; mass is normalized, and therefore unitless. In the presence of acetate, both strains are able to sustain growth as indicated by an increase in mass over time, with the wild type strain outperforming the sta6 mutant (overall biomass in black line). Without acetate (only CO₂), the model predicts much slower growth for the WT strain and no growth for the sta6. B) Predicted lipid content for WT and sta6 strains in constant light (solid bars) and in diurnal light (lined bars). C) Phenotype of individual gene knockouts for every gene in the metabolic model, once again using unitless normalized mass.

7. Figure 3 legend, ...”In the presence of acetate, both strains grow well, “ what do you mean with grow well? Maybe a more accurate word here?

Thanks for raising this point. Both strains gain biomass over time when grown on acetate; this has been clarified in lines 981-983:

In the presence of acetate, both strains are able to sustain growth as indicated by an increase in mass over time, with the wild type strain outperforming the sta6 mutant (overall biomass in black line).

8. Figure 4 legend. Would be good to have a description of the figure here.

Thank you for noting this. We’ve added an image description to the caption (lines 991-1001) and reproduced it below; we also exported it as a higher resolution pdf as well.

Process takes the form of a flowchart; one information stream flows from “Discrete Transient Systems Biology Data” into “Dynamic Biomass Setpoint Integration” and from there into “Mines Wendian High Performance Computing” via “Time Point Evaluation and Biomass Fraction Goal Seeking”. Another flows from “Curated Potential Transcriptomic Models” to “Transcriptomic Data Fit to Best Fit Continuous Function” and from there to both the endpoint of “Whole Transcriptome Expression Clustering” via “Transcript-Transcript Correlation Scoring” and to “Mines Wendian High Performance Computing”, drawing from “Genome Scale Metabolic Model” along the way via “Eflux2-Based Bound Adjustment at Each Time Point”. “Mines Wendian High Performance Computing” flows into “Transient Flux Maps and Evaluation of Growth and Knockout Viability”, the final end point.

9. Did you perform any statistical analysis to validate your results?

Thank you for asking. We compared the STA6 model to experimental growth results under different conditions to validate our results, culminating most notably the prediction of a fatality. We are not able to do any other statistical analysis on gene knockout analysis because the data for *Chlamy* is not available. This is something our lab plans to do in the future.

May 24, 2022

Prof. Nanette R Boyle
Colorado School of Mines
Chemical & Biological Engineering
1613 Illinois St
Golden, CO 80401

Re: mSystems00176-22R1 (**Rhythm of The Night (and Day): Predictive metabolic modeling of diurnal growth in *Chlamydomonas***)

Dear Prof. Nanette R Boyle:

Thank you for addressing the reviewers comments with rigor. They were both satisfied. I note that there are several text errors in the paper: for example, L114 "associated expression profiles in Error! Reference source not found" Please address these in the proofs.

Your manuscript has been accepted, and I am forwarding it to the ASM Journals Department for publication. For your reference, ASM Journals' address is given below. Before it can be scheduled for publication, your manuscript will be checked by the mSystems production staff to make sure that all elements meet the technical requirements for publication. They will contact you if anything needs to be revised before copyediting and production can begin. Otherwise, you will be notified when your proofs are ready to be viewed.

Publication Fees:

We recognize that the video files can become quite large, and so to avoid quality loss ASM suggests sending the video file via <https://www.wetransfer.com/>. When you have a final version of the video and the still ready to share, please send it to mSystems staff at mSystems@asmusa.org.

For mSystems research articles, if you would like to submit an image for consideration as the Featured Image for an issue, please contact mSystems staff at mSystems@asmusa.org.

Sincerely,

Hans Bernstein
Editor, mSystems

Journals Department
Supplemental Material 1: Accept
Supplemental Material: Accept
Supplemental Material 5: Accept
Supplemental Material 4: Accept
Supplemental Material: Accept